# *Trichoderma longibrachiatum* TG1 Colonization and Signal Pathway in Alleviating Salinity and *Fusarium pseudograminearum* Stress in Wheat

**DOI:** 10.3390/ijms26094018

**Published:** 2025-04-24

**Authors:** Solomon Boamah, Shuwu Zhang, Bingliang Xu, Na Zhu, Enchen Li

**Affiliations:** 1State Key Laboratory of Aridland Crop Science, College of Plant Protection, Gansu Agricultural University, Lanzhou 730070, China; boamahsolomon15@gmail.com (S.B.); 15693131025@163.com (N.Z.); liec188@163.com (E.L.); 2Biocontrol Engineering Laboratory of Crop Diseases and Pests of Gansu Province, Lanzhou 730070, China

**Keywords:** *Trichoderma longibrachiatum*, wheat seedlings, *Fusarium pseudograminearum*, green fluorescent protein, phytohormones, mycoparasitism, stress response

## Abstract

*Fusarium pseudograminearum* (Fp) and soil salinity are two types of stress that interact in complex ways, potentially leading to more severe consequences on wheat growth and productivity. However, little is known about the colonization efficiency and the signal pathways of the beneficial *Trichoderma longibrachiatum* TG1 (TG1) in controlling wheat Fusarium crown rot caused by Fp, and enhancing wheat seedling growth under combined salinity and Fp stresses. Therefore, the present study aims to determine the colonization, phytohormone profile, and signaling pathway in TG1-treated wheat seedlings under salinity and Fp stresses. In a dual culture assay, TG1 exhibited a mycoparasitic effect on Fp growth by coiling, conidial attachment, and parasitism observed under fluorescent microscopy. In addition, TG1 colonized the outermost layers of the wheat seedling roots with biomass consisting of conidia and hyphae. Under 100 mM NaCl stress, the combined TG1+Fp-treated seedlings recorded a control efficacy of 47.01% for the wheat crown rot disease compared with Fp-alone-treated seedlings. The contents of indole-3-acetic acid (IAA), gibberellic acid (GA_3_), abscisic acid (ABA) and jasmonic acid (JA) significantly increased by 72.16%, 86.91%, 20.04%, and 50.40%, respectively, in the combined TG1+Fp treatments, whereas the ethylene (ET) content decreased by 39.07% compared with Fp alone at day 14; and 5.07 and 2.78-fold increases in the expression of salicylic acid (SA) signaling pathway genes, such as pathogenesis-related protein 1 (*PR1*) and isochorismate synthase 1 (*ICS1*) genes were recorded respectively, in the combined TG1+Fp-treated seedlings compared with the control at day 14.

## 1. Introduction

Soil salinization is recognized as one of the most serious threats to agricultural production, affecting more than one billion hectares of land in over 100 countries worldwide [1]. Pathogens can also have devastating effects on plants, impacting their growth, development, and productivity. Infected plants may exhibit morphological changes such as lesions, necrosis, wilting, yellowing, stunted growth, and deformation [2]. Physiologically, pathogens can disrupt plant photosynthesis, nutrient uptake, and hormone balances, leading to growth and productivity reduction [3]. Wheat is a major cereal crop in the North China Plain. Recently, it has faced on the negative effects of biotic and abiotic stresses [4,5]. Saddiq et al. reported that soil salinization can cause a number of negative effects, including physiological and biochemical changes in plants, which manifests as a reduction in plant biomass and crop yield [6]. In addition, wheat crown rot is a serious soil-borne disease that mainly caused by *Fusarium pseudograminearum* (Fp) in wheat growing regions worldwide, and leading to significant yield losses up to 100% in Australia and 65% in North America under favorable conditions [7]. In China, an outbreak resulted in a 70.6% yield losses, and with mycotoxin contamination further complicated the issue in 2019 [8]. Moreover, the previous studies have reported that the interaction of salt stress and fungi infection, which causes detrimental effects on wheat [9,10], such as the *F. graminearum* infection can decrease the seed dry weights, germination, and seed vigor in both resistant and susceptible wheat cultivars [11]. In our previous study, we found that different NaCl concentrations and Fp stresses decreased the emergence parameters of wheat seedlings growth and development [9].

Plants, as sessile organisms, employ various mechanisms mediated by enzymes, proteins, or genes as their first line of defense against stress [12]. Heat shock proteins (HSPs) and genes play the vital role in plant defense by stabilizing proteins and cellular structures, helping plants cope with stressful conditions [13]. For instance, cellular stress triggers the appearance of denatured proteins and polypeptides, which in turn upregulate the expression of HSPs, leading to a significant increase in their levels within the cell. While HSPs are primarily induced by heat shock, they can also be upregulated in response to various other stresses such as salinity, drought, oxidation, and a wide range of chemicals and contaminants [14]. In a previous study, the wheat hybrid Jinan 177 and its salt-resistant hybrid protein profiling showed that HSPs and chaperones were highly induced under salt stress [15]. Additionally, mitochondrial *HSP70* gene was upregulated in rice roots under salt stress, possibly regulating programmed cell death [16]. However, there are less studies on the effect of beneficial biocontrol agents such as *T. longibrachiatum* TG1 (TG1) on *HSPs* genes expression in wheats under dual stress such as salinity and Fp stresses.

Furthermore, plant phytohormones play the pivotal role in mediating and alleviating stress responses, allowing plants to cope with both abiotic and biotic stresses. There are several classes of phytohormone, including auxins, cytokinins, gibberellins (GAs), abscisic acid (ABA), ethylene (ET), salicylic acid (SA), and jasmonic acid (JA), among others. SA, JA, ABA, and ET are four key signaling molecules that play important roles in regulating plant responses to various environmental stresses [17]. Among the different signaling molecules, *Trichoderma* spp. have been proven to increase the SA contents in wheat [9,18], maize, and rice under salinity stress [19]. In addition, SA plays a vital role in plant defense responses against biotic stress [20] through cross-talks with GA, JA, ABA, and ET. The isochorismate synthase (ICS) and phenylalanine ammonia-lyase (PAL) pathways are two distinct but interconnected pathways that play important roles in the biosynthesis of SA [21]. The ICS-SA pathway involves the conversion of chorismate, an intermediate in the shikimate pathway, into isochorismate by the enzyme ICS [21]. Isochorismate is then converted into SA by the action of the enzyme salicylate synthase (SAS). However, the *Trichoderma* underlying signal pathway in combined salinity and pathogen stresses in wheat seedlings is still unknown.

Our previous studies revealed that TG1 has a higher potential in promoting wheat seedling growth and increasing endogenous SA content under different concentrations of NaCl and Fp stresses [9,22]. However, the previous studies failed to determine the colonization, phytohormone profile, and signaling pathway in TG1-treated wheat seedlings under salinity and Fp stresses. Therefore, this study aims to determine the colonization, phytohormone profile, and signaling pathway in TG1-treated wheat seedlings under salinity and Fp stresses.

## 2. Results

### 2.1. TG1-Green Fluorescence Protein (GFP) Transformation

The TG1 transformant exhibited strong green fluorescence in hyphae (Figure 1D) and spores (Figure 1F) under a fluorescence microscope which corresponded to the bright appearance observed under bright field without fluorescence (Figure 1C,E). The spore concentration and colony growth diameter of the wild-type and transformed strains showed no statistical difference. The transformed strain GFP-TG1 exhibited colony growth diameter and spore concentration of 8.45 cm and 34.35 × 10^8^ spores/mL (Figure 1B) at 5 days after inoculation, compared with the wild-type, with 8.50 cm and 35.47 × 10^8^ spores/mL (Figure 1A).

### 2.2. Mycoparasitic Effect and Root Colonization of TG1

In the dual culture assay, TG1 demonstrated a mycoparasitic effect on the growth of Fp through conidial attachment to Fp hyphae (Figure 2A), and then the TG1 hyphae overgrew on the Fp hyphae and initiated parasitism (Figure 2B). Thereafter, the TG1 hyphae coiled, entangled, or wrapped around the Fp hyphae (Figure 2C). Additionally, in plant treatments, TG1 colonized the outermost layers of the wheat seedling roots with biomass consisting of conidia and hyphae. Under the fluorescence microscope, TG1 conidia were observed firmly attached to the root surface (Figure 2E), in contrast with the control-treated seedling (sterile water) roots (Figure 2D). Similarly, TG1 hyphal colonization and extension were clearly visualized on the root surface (Figure 2F), compared with the control.

### 2.3. Efficacy of TG1 Transformant in Controlling Fusarium Crown Rot Disease Under Salinity Stress

The disease index of Fusarium crown rot of wheat seedling that treated with TG1 wild-type or TG1-GFP transformant was lower than Fp treatment with or without salt stress. The disease index and control efficacy for the wild-type WT-TG1+Fp and transformant GFP-TG1+Fp-treated seedlings were not statistically significant. Under 0 mM NaCl stress, the control efficacies of WT-TG1+Fp and GFP-TG1+Fp treatments were 59.48% and 56.71%, respectively, compared with Fp alone. Under 100 mM NaCl stress, control efficacies of 47.01% and 41.42% were recorded for the WT-TG1+Fp and GFP-TG1+Fp treatments, respectively, compared with Fp alone (Table 1).

### 2.4. Phytohormone Contents in TG1-Treated Wheat Seedlings Under Salinity and Fp Stresses

The TG1 treatments significantly increased the contents of IAA, GA_3_, ABA, and JA, and decreased the ET content of the wheat seedlings across the different days (7, 14, 21, and 28), respectively, compared with the sterile water-treated seedlings with or without salinity stress (Figure 3A–E). Similarly, the combined TG1+Fp-treated seedlings also increased the IAA, GA_3_, ABA, and JA contents and decreased the ET content of wheat seedlings across the different days, respectively, compared with Fp-alone-treated seedlings with or without salinity stress. Comparatively, the predominant increase in IAA (Figure 3A), GA_3_ (Figure 3B), ABA (Figure 3C), and JA (Figure 3D) contents and the decrease in ET (Figure 3E) content were observed at day 14. Under 0 mM NaCl stress, the IAA, GA_3_, ABA, and JA contents of TG1-treated seedlings increased by 30.45%, 33.37%, 39.59%, and 74.72%, and ET content decreased by 22.71% at day 14, respectively, compared with the control (sterile water). Similarly, the IAA, GA_3_, ABA, and JA contents of the combined TG1+Fp-treated seedlings increased by 50.94%, 72.89%, 11.12%, and 53.03%, and the ET content decreased by 35.08% under 0 mM NaCl stress, respectively, compared with Fp alone at day 14. Under 100 mM NaCl stress, the IAA, GA_3_, ABA, and JA contents of TG1-treated seedlings increased by 25.50%, 17.80%, 32.26%, and 92.25%, and the ET content decreased by 33.86% at day 14, respectively, compared with the control (sterile water). Similarly, the IAA, GA_3_, ABA, and JA contents of TG1+Fp-treated seedlings increased by 72.16%, 86.91%, 20.04%, and 50.40%, and the ET content decreased by 39.07%, respectively, compared with the Fp alone at day 14 under 100 mM NaCl stress.

### 2.5. SA Signaling Pathway Genes Expression in TG1-Treated Wheat Seedlings Under Salinity and Fp Stresses

NaCl stress significantly affected the transcription levels of *ICS1* and pathogenesis-related 1 (*PR1*) genes across the different days (7, 14, 21, and 28), respectively, compared with the control (sterile water). Comparatively, a significant increase in both *ICS1* and *PR1* genes transcript levels were observed in TG1, TG1+Fp, and Fp-treated seedlings across the different days compared with the control (sterile water) (Figure 4A, B). The predominant increase in the *ICS1* and *PR1* genes transcript levels were observed at day 14 under salinity and Fp stresses. Under 0 mM NaCl stress, the *PR1* gene transcript levels of TG1, TG1+Fp, and Fp-treated seedlings increased by 4.76, 5.45, and 2.42-fold at day 14, respectively, compared with the control (sterile water) (Figure 4A). Similarly, the *ICS1* gene transcript levels of TG1, TG1+Fp, and Fp-treated seedlings increased by 3.00, 3.49, and 2.75-fold, respectively, compared with the control (sterile water) at day 14 (Figure 4B). Under 100 mM NaCl stress, the *PR1* gene transcript levels of TG1, TG1+Fp, and Fp-treated seedlings increased by 4.35, 5.07, and 1.03-fold, respectively, compared with the control (sterile water) at day 14 (Figure 4A). Similarly, the *ICS1* gene transcript levels of TG1, TG1+Fp, and Fp-treated seedlings increased by 2.64, 2.78, and 1.07-fold at day 14, respectively, compared with the control (sterile water) (Figure 4B).

### 2.6. Effect of TG1 on HSP70 Genes Expression in Wheat Seedlings Under Fp and Salinity Stresses

The effect of TG1 on the transcript levels of *HSP70* genes in wheat seedlings were evaluated under salinity and Fp stresses at day 14 after treatment. The results showed that all transcript levels of *HSP70-1* to *HSP70-14* genes were significantly upregulated and increased for the TG1, TG1+Fp, and Fp-treated seedlings with or without salinity stress compared with the control (sterile water). Under 0 mM NaCl stress, the transcript levels of *HSP70-1* to *HSP70-14* genes increased from 2.25 to 2.30-fold for the TG1-treated seedlings, respectively, compared with the control (sterile water). Similarly, under 100 mM NaCl stress, the transcript levels of *HSP70-1* to *HSP70-14* genes for the TG1-treated seedlings increased from 1.46 to 1.58-fold, respectively, compared with the control (sterile water). Compared with Fp-alone treatments, the combined TG1+Fp treatments significantly increased the transcript levels of *HSP70-1* to *HSP70-14* genes from 1.57 to 1.46-fold, respectively, under 0 mM NaCl stress. Similarly, TG1+Fp treatments increased the transcript levels of all *HSP70-1* to *HSP70-14* genes from 1.12 to 1.44-fold, respectively, under 100 mM NaCl stress, where *HSP70-7* and *HSP70-8* showed the highest transcript level increases of 2.14 and 5.87-folds (Figure 5).

### 2.7. Resistance Genes in TG1-Treated Wheat Seedlings Under Salinity and Fp Stresses

The transcript levels of defense-related genes (nonexpressor of pathogenesis-related 1 (*NPR1)*, enhanced disease susceptibility (*EDS*), phytoalexin deficient 4 (*PAD4*), salicylic acid induction-deficient 2 (*SID2*), lipid transfer protein (*LTP*), and WRKY transcription factor (*WRKY*)), JA-associated genes (lipoxygenase (*LOX*), allene oxide synthase (*AOS*), acyl-CoA oxidase (*ACX*), allene oxide cyclase (*AOC*)) and ET-associated genes (1-aminocyclopropane-1-carboxylate synthase (*ACS*), ethylene response factor 1 (*ERF1*), ethylene insensitive 2 (*EIN2*), ethylene insensitive 3 (*EIN3*)) were analyzed for the TG1-treated seedlings under salinity and Fp stresses at day 14. Under 0 mM NaCl stress, the transcript levels of *NPR1*, *EDS*, *PAD4*, *SID2*, *LTP,* and *WRKY* genes increased by 3.59, 2.16, 3.22, 3.47, 5.88, and 3.79-fold at day 14 for the TG1 treated seedlings, respectively, compared with the control (sterile water). Similarly, they increased by 2.56, 1.12, 1.32, 1.91, 2.19, and 2.40-fold under 100 mM NaCl stress. Additionally, the transcript levels of *NPR1*, *EDS*, *PAD4*, *SID2*, *LTP,* and *WRKY* genes increased by 2.20, 1.94, 1.70, 1.63, 2.80, and 4.60-fold for the combined TG1+Fp-treated seedlings under 0 mM NaCl stress, respectively, compared with Fp-alone treatment. Similarly, they increased by 2.05, 2.51, 3.61, 3.51, 3.05, and 2.34-fold under 100 mM NaCl stress (Figure 6).

Under 0 mM NaCl stress, the transcript levels of *LOX*, *AOS*, *ACX*, and *AOC* genes increased by 4.31, 5.12, 4.39, and 3.77-fold at day 14 for the TG1-treated seedlings, respectively, compared with the control (sterile water). Similarly, the transcript levels of TG1-treated seedlings *LOX*, *AOS*, *ACX*, and *AOC* genes increased by 2.27, 1.76, 2.61, and 2.60-fold under 100 mM NaCl stress, respectively, compared with the control (sterile water). The combined TG1+Fp-treated seedlings *LOX*, *AOS*, *ACX*, and *AOC* genes transcript levels increased by 1.07, 1.29, 1.36, and 1.71-fold, respectively, compared with Fp-alone treatment under 0 mM NaCl stress at day 14. Similarly, under 100 mM NaCl stress, the combined TG1+Fp-treated seedlings *LOX*, *AOS*, *ACX*, and *AOC* genes transcript levels increased by 3.32, 2.48, 1.20, and 1.35-fold, respectively, compared with Fp-alone treatment at day 14 (Figure 6).

Furthermore, under 0 mM NaCl stress, TG1-treated seedlings *ACS2*, *ERF1*, *EIN2*, and *EIN3* genes transcript levels decreased by 1.27, 1.11, 1.15, and 1.31-fold, respectively, compared with the control (sterile water) at day 14. Similarly, under 100 mM NaCl stress, TG1-treated seedlings *ACS2*, *ERF1*, *EIN2*, and *EIN3* genes transcript levels decreased by 1.33, 4.99, 3.71, and 1.45-fold, respectively, compared with the control (sterile water). The combined TG1+Fp-treated seedlings *ACS2*, *ERF1*, *EIN2*, and *EIN3* genes transcript levels decreased by 1.03, 2.05, 1.22, and 1.41-fold, respectively, compared with Fp-alone treatment under 0 mM NaCl stress at day 14. Similarly, under 100 mM NaCl stress, the combined TG1+Fp-treated seedlings *ACS2*, *ERF1*, *EIN2*, and *EIN3* genes transcript levels decreased by 1.66, 1.14, 1.35, and 1.32-fold, respectively, compared with Fp-alone treatment at day 14 (Figure 6).

## 3. Discussion

In this study, the highly efficient TG1 strain exhibited strong green fluorescence, and the transformant possessed the same biological characteristics as the wild-type strain. In the dual culture assay, the TG1 strain initiated a mycoparasitic effect on Fp by coiling, conidial attachment, and parasitism of Fp hyphae. This mycoparasitic action helps reduce the severity of fungal disease caused by pathogen like Fp, which causes crown rot, especially when environmental stress, such as salinity, make plants more vulnerable to infections. Our findings are consistent with Hasan et al. [23], who reported that *Trichoderma* and *Clonostachys rosea* mycoparasites efficiently overgrow and kill the fungal pathogen (*Botrytis cinerea*) by using infection structures and applying lytic enzymes such as cell wall-degrading enzymes (CWDEs) and toxic metabolites during interaction.

In previous studies, the co-inoculation of *Aspergillus niger* and *T. harzianum* alleviated the deleterious effects of salt stress in wheat seedlings through the solubilization of phosphorus and joint production of indoleacetic acid [24]; Sánchez-Montesinos et al. [25] reported that with the increase in sodium chloride concentrations, three isolates of *Trichoderma* showed antagonistic activity against *Pythium ultimum* on melon seedlings by reducing the incidence of the disease. However, in this study, the inoculation of TG1 alone decreased the deleterious effects of salt stress on wheat seedlings and the combined TG1+Fp treatment induced resistance to Fp by decreasing the disease index. In addition, Ruano-Rosa et al. [26] reported that the visualization of the green fluorescence protein labeled *T. harzianum* CECT 2413 (Th-GFP) biomass showed the presence of pre-germinated conidia on olive roots. Fluorescent hyphae were minimal or absent, with conidia and chlamydospores firmly attached to the root epidermis, showing little change at 14 DAI. In this study, fluorescence microscopy revealed the presence of conidia and hyphae of TG1-GFP on the surface of the wheat seedling roots at 14 DAI. The abundant TG1 conidia and hyphae on the seedling root surfaces defined their colonization potential. 

The establishment of microbial symbioses to promote plant growth and nutrient acquisition by beneficial microbes have been correlated with the biosynthesis of plant growth regulators and phytohormones [27]. A previous study has shown that a higher synthesis of IAA in cucumber by a putative mutant of *T. harzianum* enhanced the colonization of the rhizosphere, rhizoplane, roots, and stems [28]. Similarly, in this current study, the co-inoculation of TG1 and Fp increased the IAA, GA_3_, JA, and ABA contents and decreased the ET content of the wheat seedlings under the combined stress. The increase in phytohormones and regulation of the ethylene content induced plant growth, salinity tolerance, and resistance. In support of our findings, it was reported that *T. asperellum* also releases ABA, together with IAA and GA, into the culture medium, and its application to cucumber promoted seedling growth and alleviated the effects of salt stress [29]. Again, it was reported that ET regulates plant growth, development, and senescence, and it is well established that low ET concentrations in the root zone correspond to higher shoot growth [30].

From a previous study, it was reported that biotrophic and necrotrophic pathogens are generally two types of pathogens that activate the SA signaling pathway [31]. The biotrophic pathogens stimulate the transcription of *NPR1*. This, in turn, triggers the activation and accumulation of SA-associated genes, such as *PR1*, *PR2*, and *PR5*, both locally and systematically, leading to systemic acquired resistance (SAR) [31]. Similarly, in this study, the combined TG1+Fp treatment can lead to SAR, characterized by elevating the transcript levels of *PR1* and *ICS1* genes, which are genes of the SA pathway. Additionally, the exposure of tomato species to both high and low temperatures, along with infection by *Phytophthora infestans*, led to the upregulation of *HSP70* genes and the increased synthesis of HSP70 proteins [32]. In this current study, the TG1 alone and combined TG1+Fp inoculation increased the transcript levels of *HSP70* genes compared with the control under the combined stress. Similar findings were observed in *Arabidopsis* HSPs, along with PR proteins being more effective against *Rhizoctonia solani* [33]; many HSPs are induced and upregulated in saline stress situation like *HSP70* genes expression in rice seedlings [34], wheat [35], and *HSP70-9* to *12* and *HSP70-*33 genes expression in poplar [36].

In addition, another study identified genes and proteins involved in the beneficial interaction between *Trichoderma* and various plant species, both in roots and leaves during induced systemic resistance (ISR) [37]. Tomato seedlings inoculated with *T. parareesei* T6 significantly increased the transcript level of JA gene (*LOX*) 24 h after inoculation under NaCl stress in comparison with the control (sterile water) [38]. However, in this study, compared with the control (sterile water), TG1 alone or TG1+Fp treatment increased the transcript levels of JA-associated gene such as *LOX* at 14 days after inoculation with or without NaCl stress. Based on previous studies, several ET key genes, including *EIN2*, *EIN3*, and *ERF1,* have been assumed to be involved in fungal infections and salinity stress [39,40,41]. We hypothesized that *EIN2*, *EIN3*, and *ERF1* genes may be master modulators of wheat seedling immunity against Fp infection and salinity stress. Therefore, we determined that these genes functions are involved in the defense responses of wheat seedlings against Fp and salinity stresses. From our findings, salinity stress and Fp infection induced and increased the transcript levels of *EIN2*, *EIN3*, and *ERF1* genes in wheat seedlings that treated with sterile water. This finding confirmed that these key genes function as ET signaling and defense by providing immunity against salinity stress and Fp attacks. It has been demonstrated that a constitutive expression of *ERF1* gene, a downstream component of the ET signaling pathway, increases *Arabidopsis* resistance to *B. cinerea* and *Plectosphaerella cucumerina* [42]. Solano et al. [43] also found that *EIN3* directly regulates *ERF1* gene expression by binding to a primary ethylene response element present in the promoter of *ERF1*. Xu et al. [44] reported that the transcription of the wheat *TaERF1* gene was induced by salinity and *Blumeria graminis* f. sp. *tritici*, suggesting that *ERF1* gene might play a role in distinct areas under biotic and abiotic stresses. Similarly, Zhao et al. [45] reported that, when salt stress diminishes, *ERF1* gene expression decreases, thus alleviating the inhibition of lateral root emergence caused by the abnormal distribution of high auxin accumulation and promoting *Arabidopsis* root growth. Wang et al. [46] reported that transcriptomic *ERF*-related genes (*EIL1*, *ERF3*, *ERF54*, *ERF53*, *ERF6*, *ERF9*, and *ERF7*) were downregulated in *Arabidopsis* seedlings after inoculated with *T. asperellum* or *T. virid*e under salt stress. This finding indicates that the inoculation with *T. asperellum* and *T. viride* decreased ethylene synthesis, thus improving the tolerance of seedlings to salt stress. In addition, it was reported that the ET signaling regulates plant growth and development, and *EIN2*, a key gene in the ET signaling pathway, is required for salt tolerance [39]. We found that the plant growth-promoting TG1 strain decreased the transcript levels of *EIN2* gene under salinity and Fp stresses. In consistent with our findings, Rubio et al. [47] reported that the application of *T. harzianum* T34 decreased the expression level of *EIN2* gene expression under salt stress or not in comparison with the normal tomato plants (control), while the salt stress alone increased the expression level of the *EIN2* gene. 

In addition, it was reported that, plants that over-accumulate *EIN3* are compromised in pathogen-associated molecular pattern (PAMP) defenses and exhibited enhanced disease susceptibility to *Pseudomonas syringae*, and high transcription levels of *EIN3* gene negatively regulates innate immunity in *Arabidopsis* [48]. In this current study, TG1 alone or combined TG1+Fp decreased the transcript levels of *EIN3* gene, negatively regulating ethylene synthesis and promoting seedling growth, salinity tolerance, and resistance to Fp. In comparison with the previous study, the transcription level of *EIN3* gene in onion was downregulated at day 1 after inoculation with the combined *T. asperellum* and *Sclerotium cepivorum*, whereas it increased at 21 days [49]. In contrast to our study, TG1 decreased the transcription level of *EIN3* gene in wheat seedlings under salinity and Fp stresses at day 14. This indicates that TG1 modulates the expression of some key ET genes by suppressing the transcription of their repressors under combined abiotic and biotic stresses.

Similarly, it was reported that, in ethylene biosynthesis, ACO and ACS are key rate-limiting enzymes. *ERFs* can interact with the *ACS2/5* genes promoter to inhibit their expression, negatively regulating ethylene synthesis [46]. In addition, the expression of *ACS6*, *ACO6*, *ACS10*, *ACS10,* and *ACS11* genes was decreased after inoculation with *T. viride* under salinity stress [46]. Consistent with our findings, the TG1 strain alone or in combination with Fp reduced ethylene synthesis in wheat seedlings by lowering the transcript levels of *ACS2* gene under combined salinity and Fp stresses. These results suggest that inoculation with the TG1 strain, which has ACCD activity, promotes plant growth by decreasing ethylene content, thereby alleviating the negative effects of both Fp and salinity stresses.

## 4. Materials and Methods

### 4.1. GFP-TG1 Strain Transformation and Fungal Inoculum Preparation

The fungi strains (*Trichoderma longibrachiatum* (TG1) and *Fusarium pseudograminearum* (Fp)) were obtained from the Laboratory of Plant Pathology, Gansu Agricultural University. TG1 and Fp were cultured on PDA media in Petri dishes for 7 and 14 days at 25 °C, respectively. Construction of the GFP-labeled strain TG1 was performed following the method of Hasan et al. [23], where TG1 spores were prepared and cultured in potato dextrose broth (PDB) at 180 rpm and 28 °C overnight. Protoplasts were filtered with three layers of lens tissue paper, suspended in sorbitol–Tris–calcium (STC) buffer and stored on ice. Polyethylene glycol CaCl_2_-mediated transformation of GFP was conducted. The pBR322 plasmid was propagated in *Escherichia coli* DH5α and cultured on Luria–Bertani (LB) medium with 100 µg/mL ampicillin, and the plasmid DNA was extracted using a TIANprep Rapid Mini Plasmid Kit (Tiangen Biotech Co., Ltd., Beijing, China). The protoplast suspension (200 µL) was mixed with 5 µg plasmid DNA and incubated on ice for 20 min, and then 1.25 mL of 40% polyethylene glycol–Tris–calcium solution was added. After 20 min, the mixture was poured into the LB medium containing ampicillin, followed by incubating at 28 °C. Transformants were obtained after 5 days. After four generations, stable transformants were checked under fluorescence microscopy (Zeiss™ Axioscope 5, Carl Zeiss Suzhou Co., Ltd., Suzhou, China). Transformants demonstrating robust fluorescence were selected for further experiment. The spore suspensions of TG1 and Fp were prepared according to the method of Zhang et al. [50]. The spore suspension of TG1 (1.0 × 10^8^ spores/mL) and Fp (5 × 10^8^ spores/mL) were quantified and stored at 4 °C.

### 4.2. Mycoparasitic Effect of GFP-TG1 on Fp and Determination Its Root Colonization In Vitro Conditions

The mycoparasitic effect of GFP-TG1 on Fp was observed using a dual plate culture method following the procedures of Yassin et al. [51]. Briefly, the cellophane fragments in the interaction zone were examined after following contact (overgrowth) at 4 days. Interaction between the fungi was observed under fluorescence microscopy (Zeiss™ Axioscope 5, Carl Zeiss Suzhou Co., Ltd., Suzhou, China). 

Wheat seeds with equal sizes were surface sterilized with 1% NaOCl solution for 10 min and rinsed with sterile water for six times after disinfection. Thereafter, wheat seeds were soaked in GFP-TG1 spore suspension at a concentration of 1.0 × 10^8^ spores/mL and sterile water (control) for 12 h, respectively. The treatment-seeds were air-dried overnight under aseptic conditions, and then exposed to soil. Each treatment and control having three replicates, and each replicate with one plastic pot (9 cm in diameter and 10.5 cm in depth, containing 0.54 kg of sterile soil). Twenty uniform seeds were sown 1 cm deep in soil and lightly covered. The wheat seedlings were maintained in a growth chamber at 25/20 °C day/night temperature under a 16/8 h light/dark photoperiod. Following periodic irrigation, seedlings were harvested at 14 days after treatment. For the observation of root colonization by TG1, wheat seedling roots in each treatment and control were collected, thoroughly washed with sterile water and excised for the observation under the fluorescence microscopy. 

### 4.3. Efficacy of GFP-TG1 in Controlling Fusarium Crown Rot Disease Under Salinity Stress

The sterile wheat seeds were soaked in (i) Fp spore suspension (5.0 × 10^8^ spore/mL) only for 12 h, (ii) Fp spore suspension for 12 h followed by WT-TG1 spore suspension (1 × 10^8^ spore/mL) for 12 h and (iii) Fp spore suspension for 12 h followed by GFP-TG1 spore suspension (1 × 10^8^ spore/mL) for 12 h, respectively. Thereafter, all the seeds in different treatments were air-dried and sown in pots that contained 0.54 kg of sterile soil at 0 or 100 mM NaCl concentrations. Following periodic irrigation, wheat seedlings were harvested at 14 days after treatment. Each treatment having three replicates (pots). The disease index was assessed using a 0–5 point scale (0, symptomless; 1, slightly necrotic; 2, moderately necrotic; 3, severely necrotic; 4, completely necrotic and 5, death), following the method of Zhang et al. [52].

### 4.4. Effect of TG1 on the Physiological and Molecular Parameters of Wheat Seedlings Growth and Disease Resistance Under Salinity and Fp Stresses

The sterile wheat seeds were soaked in WT-TG1 (1.0 × 10^8^ spore/mL) or Fp (5 × 10^8^ spore/mL) spore suspension for 12 h. For the combined (TG1+Fp), seeds were soaked in Fp spore suspension for 12 h followed by soaking in TG1 spore suspension for 12 h. The control seeds (CK) were soaked in sterile water for 12 h. Seeds in each treatment and control were air-dried overnight under aseptic conditions and exposed to artificial saline soil at 0 or 100 mM NaCl. The detailed treatments were grouped into eight (T1-T8), where T1 (CK+0 mM NaCl) represents wheat seeds treated with sterile water and subjected to no salt stress (0 mM NaCl), T2 (CK+100 mM NaCl) represents wheat seeds treated with sterile water and subjected to 100 mM NaCl stress, T3 (TG1+0 mM NaCl) represents wheat seeds treated with TG1 spore suspension and subjected to no salt stress (0 mM NaCl), T4 (TG1+100 mM NaCl) represents wheat seeds treated with TG1 spore suspension and subjected to 100 mM NaCl stress, T5 (TG1+Fp+0 mM NaCl) represents wheat seeds treated with TG1 and Fp spore suspension, and subjected to no salt stress (0 mM NaCl), T6 (TG1+Fp+100 mM NaCl) represents wheat seeds treated with TG1 and Fp spore suspension, and subjected to 100 mM NaCl stress, T7 (Fp+0 mM NaCl) represents wheat seeds treated with Fp spore suspension and subjected to no salt stress (0 mM NaCl), and T8 (Fp+100 mM NaCl) represents wheat seeds treated with Fp spore suspension and subjected to 100 mM NaCl stress. Each treatment having three replicates (pots), each pot containing 0.54 kg of air-dried sterile saline soil or non-saline soil. Following periodic irrigation, wheat seedlings were harvested at 7, 14, 21, and 28 days after treatment for physiological and molecular analysis.

### 4.5. Measurement of Phytohormones in TG1-Treated Wheat Seedlings Under Salinity and Fp Stresses

Fungi and sterile water-treated wheat seedlings subjected to 0 and 100 mM NaCl were harvested at different time points (7, 14, 21 and 28 days) into liquid nitrogen, freezed and grounded. 0.1 g of grounded leaves tissue was weighed into 1.5 mL microfuge tubes and extracted with 400 μL of 10% methanol containing 1% acetic acid. The supernatant was carefully removed and the pellet re-extracted with 400 μL of 10% methanol containing 1% acetic acid. Following a further 30 min incubation on ice, the extract was centrifuged and the supernatant obtained. The IAA, ABA, GA_3_, and JA contents determinations were performed by using high-pressure liquid chromatography (HPLC) (Agilent 1260 HPLC, Waters ACQUITY Arc UHPLC, Agilent Technologies Inc., Santa Clara, CA, USA) following the methods of Allasia et al. [53] and Hou et al. [54]. Briefly, HPLC was performed using (4.6 × 250 mm, 5 µm) columns at room temperature with a combination of 0.1% trifluoroacetic acid (A) and acetonitrile (B) at a flow rate of 1 mL/min. Fluorometric detection was measured at 254 nm. Individual standard compounds were prepared to determine the concentrations, and the HPLC-generated standard curve with linear equations was derived from peak areas for analysis. The content of ET was determined by Gas Chromatography using an Agilent 7080B gas chromatograph (Agilent 7080B, Santa Clara, CA, USA) under the following chromatographic conditions: PoraparkQ 15 m × 0.53 mm × 25 µm, injection port at 150 °C, split injection, ECD detector, detection temperature at 300 °C, column temperature at 45 °C, and column flow rate of 3.3 mL/min [55].

### 4.6. Extraction of Total RNA

The wheat seedlings leaves were collected at 7, 14, 21 and 28 days after treatments, and used for RNA extraction. The specific method for the samples RNA extraction was performed following the manufacturer’s protocol of the E.Z.N.A^®^ plant RNA kit (OMEGA Bio-Tek, Norcross, GA, USA). The quantity and purity of the isolated RNA were analyzed using a Nanophotometer (IMPLEN, Schatzbogen, Germany). First-strand cDNA was synthesized using a Revert AidTM First Strand cDNA Synthesis Kit (Tiangen Biotechnology, Beijing, China) following the manufacturer’s protocol. The first-strand cDNA was used for genes expression analysis using qRT-PCR.

### 4.7. Analysis of Genes Expression by qRT-PCR

The transcript levels of *ICS1* and *PR1* genes were evaluated at four different time points (7, 14, 21, and 28 days), while 14 *HSP70* genes transcript levels were evaluated at day 14. The transcription levels of *NPR1*, *EDS*, *PAD4*, *SID2*, *LTP*, *WRKY*, *LOX*, *AOS*, *ACX*, *AOC*, *ACS2*, *ERF1*, *EIN2*, and *EIN3* genes were also analyzed at day 14 after treatment. Primers used in the experiments were designed using Primer Express 5.0 software, which amplifies target genes according to the sequences of candidate genes available in NCBI wheat EST [56]. Two wheat actin genes were used as endogenous controls (Table 2). qRT-PCR was performed using 2 × M5 HiPer Real-time PCR Supermix with Low Rox (Mei5 Biotechnology, Co., Ltd., Beijing, China). Three biological replicates were used for each gene. The 2^−ΔΔCt^ method was used to measure the relative expression of the genes [57]. 

### 4.8. Statistical Analysis

IBM SPSS^®^ Statistics 27 (IBM Corp., Armonk, NY, USA) was used for statistical analysis using one-way analysis of variance. Results were expressed as mean value ± standard error (SE). *p*-values less than 0.05 were considered to be significant using Duncan’s new multiple range test.

## 5. Conclusions

TG1 exhibited a mycoparasitic effect on Fp growth by coiling, conidial attachment, and parasitism, which was observed under fluorescent microscopy. TG1 pretreatment of wheat seedlings-initiated root surface colonization and decreased the disease index of Fusarium crown rot. Meanwhile, the TG1 pretreatment increased the IAA, GA_3_, JA, and ABA contents of wheat seedlings, whereas decreased the ET content under salinity and Fp Stresses. In addition, TG1 induced salinity tolerance and resistance of wheat seedlings to Fp infection by increasing the transcript levels of wheat salinity tolerance and resistance genes expression. 

## Figures and Tables

**Figure 1 ijms-26-04018-f001:**
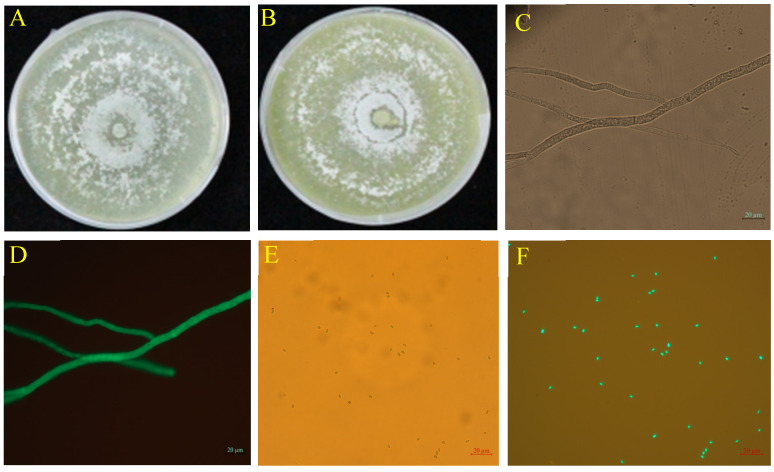
Colony morphology and microscopic characteristics of wild-type and GFP-TG1 transformant on PDA plates at 5 days after inoculation. (**A**) Wild-type TG1, (**B**) GFP-labeled TG1, (**C**) Bright field microscopic image of GFP-labeled TG1 hyphae, (**D**) Fluorescence microscopic image of GFP-labeled TG1 hyphae, (**E**) Bright field microscopic image of GFP-labeled TG1 spores, and (**F**) Fluorescence microscopic image of GFP-labeled TG1 spores.

**Figure 2 ijms-26-04018-f002:**
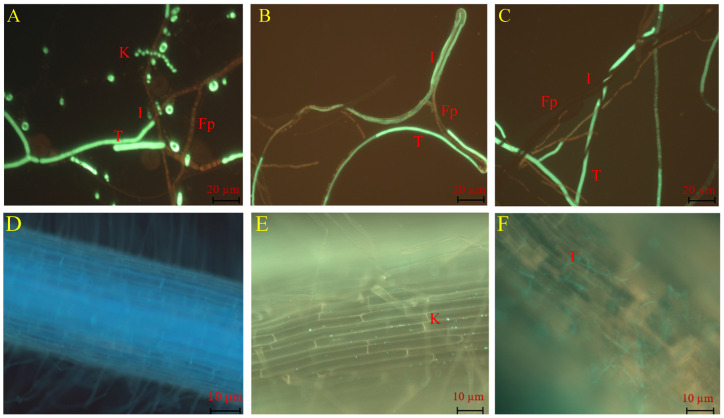
Visualization of TG1 mycoparasitic effect on Fp and wheat root colonization. Where T and K represent TG1 hyphae and conidia, respectively; Fp represents Fp hyphae, and I represent the interaction between TG1 and Fp. (**A**) Conidia attachment of Fp hyphae, (**B**) TG1 hyphae exhibited parallel and overgrowth on the Fp hyphae, (**C**) TG1 hyphae entangled, coiled, or wrapped the hyphae of Fp, (**D**) root surface of wheat seedlings treated with sterile water (control), (**E**) TG1 conidia colonization on the root surface, and (**F**) TG1 hyphae colonization on the root surface.

**Figure 3 ijms-26-04018-f003:**
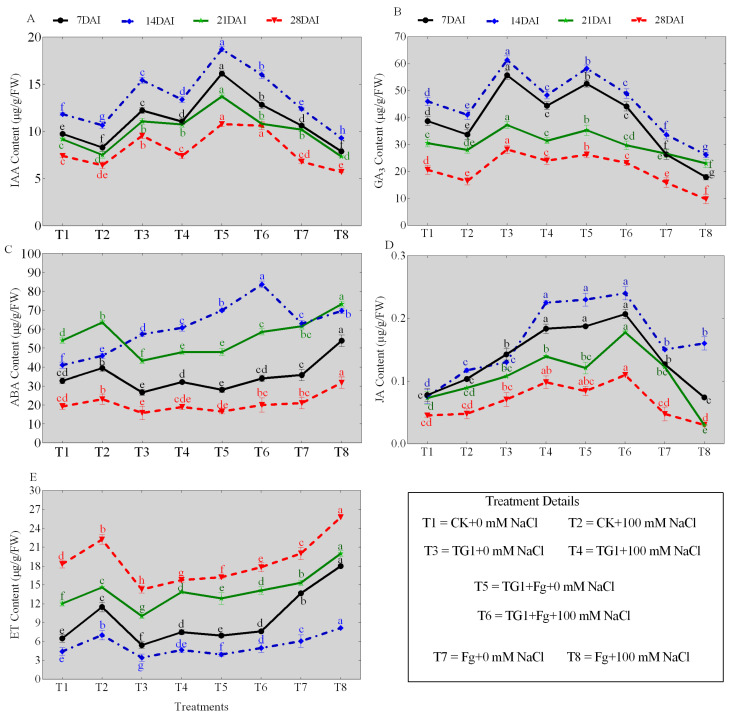
Effect of TG1 on phytohormone contents of wheat seedlings under the combined salinity and Fp stresses at different days (7, 14, 21, and 28). Where (**A**–**E**) represent the contents of IAA, GA_3_, ABA, JA, and ET, respectively. Where T1 (CK+0 mM NaCl) represents wheat seeds treated with sterile water and subjected to no salt stress (0 mM NaCl), T2 (CK+100 mM NaCl) represents wheat seeds treated with sterile water and subjected to 100 mM NaCl stress, T3 (TG1+0 mM NaCl) represents wheat seeds treated with TG1 spore suspension and subjected to no salt stress (0 mM NaCl), T4 (TG1+100 mM NaCl) represents wheat seeds treated with TG1 spore suspension and subjected to 100 mM NaCl stress, T5 (TG1+Fp+0 mM NaCl) represents wheat seeds treated with TG1 and Fp spore suspension and subjected to no salt stress (0 mM NaCl), T6 (TG1+Fp+100 mM NaCl) represents wheat seeds treated with TG1 and Fp spore suspension and subjected to 100 mM NaCl stress, T7 (Fp+ 0 mM NaCl) represents wheat seeds treated with Fp spore suspension and subjected to no salt stress (0 mM NaCl), and T8 (Fp+100 mM NaCl) represents wheat seeds treated with Fp spore suspension and subjected to 100 mM NaCl stress. Different lowercase letters indicate significant differences at *p* < 0.05 in Duncan’s multiple-range test using one-way ANOVA.

**Figure 4 ijms-26-04018-f004:**
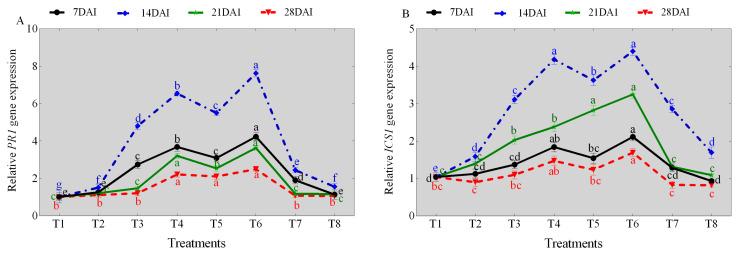
Transcription levels of (**A**) *PR1* and (**B**) *ICS1* genes for the TG1-treated wheat seedlings under salinity and Fp stresses at different days (7, 14, 21, and 28). Treatments are detailed in the footnote of Figure 3. Different lowercase letters indicate significant differences at *p* < 0.05 in Duncan’s multiple range test using one-way ANOVA.

**Figure 5 ijms-26-04018-f005:**
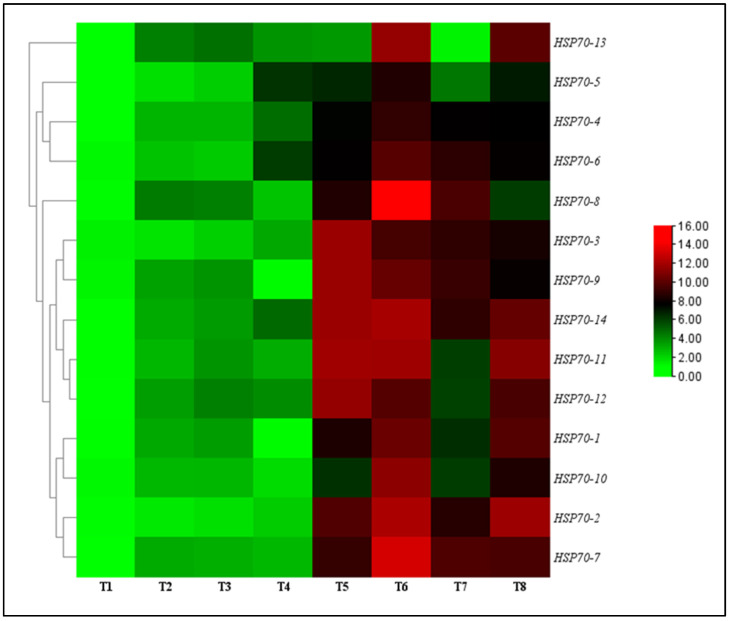
Relative transcription levels of *HSP70* genes in TG1-treated wheat seedlings under Fp and salinity stresses at day 14. The color bar is the scale of relative transcript levels of the genes. Treatments are detailed in the footnote of Figure 3.

**Figure 6 ijms-26-04018-f006:**
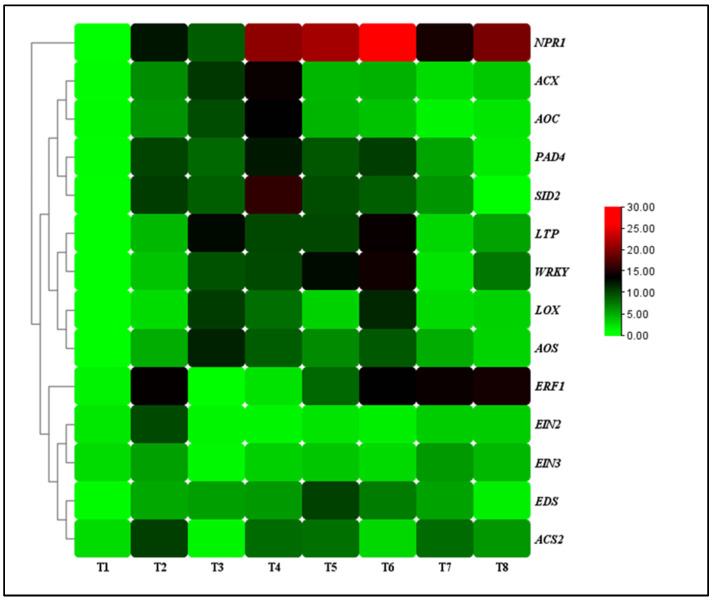
The transcript levels of resistance genes in TG1-treated wheat seedlings under Fp and salinity stresses at 14 days after treatment. The color bar is the scale of relative transcript levels of the genes. Treatments are detailed in the footnote of Figure 3.

**Table 1 ijms-26-04018-t001:** Control efficacy of TG1 and its transformant treatments on wheat Fusarium crown rot disease under salinity stress.

NaCl Concentrations (mM)	Treatments	Disease Index	Control Efficacy (%)
0	Fp	32.50 ± 0.64 ^c^	—
	WT-TG1+Fp	13.17 ± 0.24 ^d^	59.48 ± 1.45 ^a^
	GFP-TG1+Fp	14.07 ± 0.34 ^d^	56.71 ± 1.73 ^a^
100	Fp	68.64 ± 2.61 ^a^	—
	WT-TG1+Fp	36.37 ± 1.92 ^bc^	47.01 ± 4.25 ^b^
	GFP-TG1+Fp	40.21 ± 1.46 ^b^	41.42 ± 3.03 ^b^

Data are means ± standard error (SE) of three replicates. Different letters in a column indicate significant differences at *p* < 0.05.

**Table 2 ijms-26-04018-t002:** List of primes used in this study.

Genes ID	Genes Name	Primers Sequence (5′-3′)
9268489	*ICS1*	F: GGGCACCCCTCACCTCACT R: GATGAATGGGTGCGGCG
543437	*PR1*	F: TGGGTGGACGAGAAGAAGGA R: TACTAACTGTGATTGCTCCGCAG
123111177	*Hsp70* *-1*	F: TTGCTTTTGCCGAGGATGG R: GCTTTTGCCGAGGATGGTG
123180708	*Hsp70-2*	F: GAGGAAGGGTTTGACGAAGAAGT R: AAACCTACAACATCCCCGTGG
123146355	*Hsp70-3*	F: AAAGTGTCAAAGGCGATGGTAAT R: TCTTCCCCTCTTCGTTTGCG
542817	*Hsp70-4*	F: CCTACCTTCGTTTCCAGGCTAA R: AGCAAGTATGTCTCCGTACCGC
123040814	*Hsp70-5*	F: GTGGTAGGATGGGGACCTGATA R: CTATTCAGGCAGGCTTCCTCC
123091767	*Hsp70-6*	F: GATGTGGTTGTTGTTTGCGACT R: GCACCCCAGATCCATAATCG
542820	*Hsp70-7*	F: CATTGCCATTGCCCGAAAC R: GGGACAAGGACTTTGGTGGC
123079701	*Hsp70-8*	F: CCAAGGACTCCCGACGAAA R: ATCGTCCGCTCCATTGTTCTT
123141999	*Hsp70-9*	F: ACGGCAGAGCAGGTCAAAGA R: ATCGTGCTGGTGGGAGGC
123119701	*Hsp70-10*	F: TTTCGGTGCCGCTGTTCA R: GGAGGATAAGGCTACCCGTGA
123071367	*Hsp70-11*	F: ACGCACCGTCCACATCCC R: AAACAACGCCCGCAGTCTC
123170168	*Hsp70-12*	F: CTCCGAATCATCAACGAGCCT R: AACGAGCCTACCGCTGCC
123159855	*Hsp70-13*	F: GCTTTGAGGGAGGCGAGTTT R: AACGAGCCTACCGCTGCC
123157046	*Hsp70-14*	F: AAGAACAATGCTGCCATGAACC R: TTTTGGCGAGCGGGCTT
842733	*NPR1*	F: CTGCGATGCGGAAGGAGC R: GTTCCTCCCTCTTTTGCAGTGG
823964	*EDS*	F: AGACGGGGAGGTAGATGATAAGG R: CACAAATCGCATTCTCCTCTGC
123086651	*PAD4*	F: GCATTCAAGCAACGGAGGTC R: ATGTCGGTCGTACCCGCTCTA
843810	*SID2*	F: CGCCACTTGGTACAGTGGAGA R: CTCAAATCTCAACCTCCGTCGT
101261679	*LTP*	F: TAAGAACCAGCAGCCTTGCAT R: TTGCTGGACAATGTGGTGTAAGC
123121634	*WRKY*	F: TGAAGGTTGTGAAAGACGGGTAT R: CCGTCCTGCCCTGTCAAGA
123113437	*AOS*	F: CGTCATCCCCACGTTCCG R: GCAACTACAACGACGACGCC
123069729	*ACX*	F: TGGCTGGGCTGGCATAGTG R: TCTTTTCCTGTCCATGTCTGTCC
123138464	*AOC*	F: CCAAGCAGAGCCAGAACGC R: GCTCCTCTGCTTGCTTTGCTG
100313965	*LOX*	F: TCCATGACCTGATCCTTCCCTT R: GAGGCGATTATTGCTCTAGTTGG
543237	*ACS2*	F: ATCGCGTCGTGATGAGCG R: CCCACGCCTTACTACCCAGG
123087171	*EIN2*	F: GCTTGACCTGTCCCTTGTGG R: GGGTGTATGCTAGAGTCGCAGTT
123051514	*EIN3*	F: CCGCTCTTTCCACATCTCCTG R: AGGGAAAACAAGCAGAGCAACC
542960	*ERF1*	F: GCTTGCATCCATTCCCACC R: TGTGATGGGTGATGCTAATGTTG
123179877	*actin-97*	F: TCTTTCGCTACACTTGGCACAT R: GGGTGACAGTATTGCTCGCC
123185737	*actin*	F: CTAATCCCACCTCAACCCAATC R: ATGGAGTTTCTTGGGTTTACGC

## Data Availability

The data presented in this study are available on request from the corresponding author.

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
