# Peer review of "Trichoderma longibrachiatum TG1 Colonization and Signal Pathway in Alleviating Salinity and Fusarium pseudograminearum Stress in Wheat"

_ijms, 2025, doi:10.3390/ijms26094018_

Round 1

Reviewer 1 Report

Comments and Suggestions for Authors

The combined study of an abiotic (salinity) and a biotic stress (Fusarium) on wheat seedlings is a good approach to address the adversaries of climate change patterns in cropping systems. Although most of the components of the study have been reported for bacterial inoculum, it is good to see some research from a fungal perspective as well, adding to the knowledgebase.

Introduction:

Kindly correct/rewrite these sentences -

Line 44: for human consumption worldwide.

Line 47: root development. It has been estimated worldwide that salinity affects 20% of total cultivated and 33% of irrigated agricultural lands.

Line 51: It is estimated that…

Line 52: under salinity stress suffer

Line 75: The use of biocontrol agents…. Include references for the reported previous studies.

Results:

Line 154: TG1 mycelium entangled, coiled or wrapped…

In the figures, kindly change the font color of the labels. They are not as visible as they should be.

Figure 2: The legend says T and C (line 168), but the figure has T, I and K?

Line 193: colonized the seedling roots…

Figure 5 is a neat representation of the phytohormones of treatments and days response.

Line 313: Comparatively, a significant increase in both ICS1 and PR1….

Line 323: FYI – It is now well known that PR1 and PR5 are the most reliable markers for SA via Phenylalanine pathway and ICS1 is the marker for SA via shikimic acid pathway. Kindly correct the sentence and the thoughts following this.

Methods:

Correct all the SI units – mL, µL, spore/mL or mg/L or keep it consistent at spore mL⁻¹ and mg L⁻¹ etc, and other formatting issues for space between words.

Line 595, 604 and wherever mentioned: qRT-PCR.

Author Response

Reviewer 1

Open Review

Response: Thank you for your comment.

Quality of English Language

(x) The quality of English does not limit my understanding of the research.

Response: Thank you for your comment.

( ) The English could be improved to more clearly express the research.

Yes

Can be improved

Must be improved

Not applicable

Does the introduction provide sufficient background and include all relevant references?

(x)

( )

( )

( )

Response: Thank you for your comment.

Is the research design appropriate?

Response: Thank you for your comment.

(x)

( )

( )

( )

Are the methods adequately described?

Response: Thank you for your comment.

(x)

( )

( )

( )

Are the results clearly presented?

Response: Thank you for your comment.

(x)

( )

( )

( )

Are the conclusions supported by the results?

Response: Thank you for your comment.

(x)

( )

( )

( )

Comments and Suggestions for Authors

The combined study of an abiotic (salinity) and a biotic stress (Fusarium) on wheat seedlings is a good approach to address the adversaries of climate change patterns in cropping systems. Although most of the components of the study have been reported for bacterial inoculum, it is good to see some research from a fungal perspective as well, adding to the knowledgebase.

Introduction:

Kindly correct/rewrite these sentences -

Line 44: for human consumption worldwide.

Response: Thank you for your comment. The introduction has been modified and excluding that sentence.

Line 47: root development. It has been estimated worldwide that salinity affects 20% of total cultivated and 33% of irrigated agricultural lands.

Response: Thank you for your comment. The introduction has been modified and excluding that sentence.

Line 51: It is estimated that…

Response: Thank you for your comment. The introduction has been modified and excluding that sentence.

Line 52: under salinity stress suffer

Response: Thank you for your comment. The introduction has been modified and excluding that sentence.

Line 75: The use of biocontrol agents…. Include references for the reported previous studies.

Response: Thank you for your comment. The references have been added. (Line 60)

Results:

Line 154: TG1 mycelium entangled, coiled or wrapped…

Response: Thank you for your comment. This sentence has been corrected. (Line 124-125)

In the figures, kindly change the font color of the labels. They are not as visible as they should be.

Response: Thank you for your comment. The figures font colors of the labels have been changed to enhanced visibility.

Figure 2: The legend says T and C (line 168), but the figure has T, I and K?

Response: Thank you for your comment. The figure legend has been modified. (Line 149)

Line 193: colonized the seedling roots…

Response: Thank you for your comment. This sentence has been corrected. (Line 206)

Figure 5 is a neat representation of the phytohormones of treatments and days response.

Response: Thank you for your comment. The figure legend has been modified and figure number has been changed to 3. (Line 198)

Line 313: Comparatively, a significant increase in both ICS1 and PR1….

Response: Thank you for your comment. This sentence has been corrected. (Line 219)

Line 323: FYI – It is now well known that PR1 and PR5 are the most reliable markers for SA via Phenylalanine pathway and ICS1 is the marker for SA via shikimic acid pathway. Kindly correct the sentence and the thoughts following this.

Response: Thank you for your comment. This sentence has been moved to the discussion.

Methods:

Correct all the SI units – mL, µL, spore/mL or mg/L or keep it consistent at spore mL⁻¹ and mg L⁻¹ etc, and other formatting issues for space between words.

Response: Thank you for your comment. These errors have been rectified throughout the manuscript.

Line 595, 604 and wherever mentioned: qRT-PCR.

Response: Thank you for your comment. These errors have been rectified (Line 612 and 616)

Reviewer 2 Report

Comments and Suggestions for Authors

This manuscript studied the control efficiency of Trichoderma longibrachiatum against wheat Fusarium crown rot (FCR), and found that under 100 mM NaCl stress, the strain of T. longibrachiatum TG1 exhibited a control efficacy of 47.02%. The author needs to make confirmation on several points in the manuscript.

Figure 2, "Where T and C represent TG1 mycelium and conidia, respectively", I didn't find C in the figure, should it be K?

Figure 3, "(F) TG1 spore (violet arrow) colonization in the root parenchyma cells", how can spores be seen inside the tissue? Don't Trichoderma spores enter the plant root tissue only after germination? Or can the hyphae that invade the host produce spores within the plant tissue?

“(H) Dual observation of TG1 spores (violet arrow) and Fg spores (red arrow) in the root endodermis”. Similarly, how can there be two fungal spores in the root endodermis? I am skeptical about these spores appearing in the host tissue, and the author needs to confirm carefully.

Author Response

Reviewer 2

Open Review

(x) I would not like to sign my review report

Response: Thank you for your comment.
( ) I would like to sign my review report

Quality of English Language

(x) The quality of English does not limit my understanding of the research.

Response: Thank you for your comment.
( ) The English could be improved to more clearly express the research.

Yes

Can be improved

Must be improved

Not applicable

Does the introduction provide sufficient background and include all relevant references?

Response: Thank you for your comment.

(x)

( )

( )

( )

Is the research design appropriate?

Response: Thank you for your comment.

(x)

( )

( )

( )

Are the methods adequately described?

Response: Thank you for your comment.

(x)

( )

( )

( )

Are the results clearly presented?

Response: Thank you for your comment. The results presentation has been improved.

( )

(x)

( )

( )

Are the conclusions supported by the results?

Response: Thank you for your comment.

(x)

( )

( )

( )

Comments and Suggestions for Authors

This manuscript studied the control efficiency of Trichoderma longibrachiatum against wheat Fusarium crown rot (FCR), and found that under 100 mM NaCl stress, the strain of T. longibrachiatum TG1 exhibited a control efficacy of 47.02%. The author needs to make confirmation on several points in the manuscript.

Response: Thank you for your comment. The confirmation of this results has been confirmed and explained with previous published literature in the discussion (410-413).

Figure 2, "Where T and C represent TG1 mycelium and conidia, respectively", I didn't find C in the figure, should it be K?

Response: Thank you for your comment. The figure legend has been modified and corrected with K. (Line 149)

Figure 3, "(F) TG1 spore (violet arrow) colonization in the root parenchyma cells", how can spores be seen inside the tissue? Don't Trichoderma spores enter the plant root tissue only after germination? Or can the hyphae that invade the host produce spores within the plant tissue?

 Response: Thank you for your comment. This figure has been revised and removed.

“(H) Dual observation of TG1 spores (violet arrow) and Fg spores (red arrow) in the root endodermis”. Similarly, how can there be two fungal spores in the root endodermis? I am skeptical about these spores appearing in the host tissue, and the author needs to confirm carefully.

 Response: Thank you for your comment. This figure has been revised and removed.